# Do Individuals with Internet Gaming Disorder Share Personality Traits with Substance-Dependent Individuals?

**DOI:** 10.3390/ijerph19159536

**Published:** 2022-08-03

**Authors:** Julie Giustiniani, Magali Nicolier, Madeline Pascard, Caroline Masse, Pierre Vandel, Djamila Bennabi, Sophia Achab, Frédéric Mauny, Emmanuel Haffen

**Affiliations:** 1Department of Clinical Psychiatry, University Hospital of Besançon, 25000 Besançon, France; mnicolier@chu-besancon.fr (M.N.); cmasse@chu-besancon.fr (C.M.); pierre.vandel@univ-fcomte.fr (P.V.); djamila.bennabi@univ-fcomte.fr (D.B.); emmanuel.haffen@univ-fcomte.fr (E.H.); 2Clinical Investigation Center 1431-INSERM, University Hospital of Besancon, 25000 Besançon, France; 3UR 481 LINC Neurosciences and Cognition, University of Franche-Comté and Bourgogne Franche-Comté, 25000 Besançon, France; 4UMETh—Inserm Centre Investigation Clinique 1431, CHU Besançon, 2 Place Saint-Jacques, 25030 Besançon, France; madeline.pascard@gmail.com (M.P.); frederic.mauny@univ-fcomte.fr (F.M.); 5Sociological and Clinical Research Unit, Faculty of Medicine, Psychiatry Department, Geneva University, 1211 Geneva, Switzerland; sophia.achab@hcuge.ch; 6FondaMental Foundation, 94000 Créteil, France

**Keywords:** internet gaming disorder, alexithymia, impulsivity, sensation seeking, aggressiveness, substance use disorder

## Abstract

(1) Background: Internet gaming disorder (IGD) shares many similarities with substance use disorder (SUD), contributing to its recognition as an addictive disorder. Nevertheless, no study has compared IGD to other addictive disorders in terms of personality traits established as highly co-occurring with SUDs. (2) Methods: We recruited a sample of gamers (massively multiplayer online role-playing games) (MMORPGs) via online in-game forums. We compared 83 individuals with IGD (MMORPG-IGD group) to 47 former heroin addicts under methadone maintenance treatment (MMT; MMT group) with regard to alexithymia, impulsivity, sensation seeking and aggressiveness assessed through self-administered scales, being TAS-20, BIS-10, Z-SSS and BDHI, respectively. (3) Results: Our results draw a relatively similar personality profile between groups but indicate that the subject traits are generally more pronounced in the MMT cohort. The overall lesser intensity of these traits in the MMORPG-IGD group might reflect the greater variability in the severity of the IGD. (4) Conclusions: IGD shares personality traits with MMT, and intensity may be influenced by the severity of the addiction or by certain direct environmental factors, and might also influence the propensity towards one behavior rather than another.

## 1. Introduction

Video games have become the world’s bestselling entertainment [1]. A proliferation of game interfaces and the multiplication of hours spent on the internet [2,3] have been directly linked to the emergence of a new disorder. Thus, while for most consumers this activity is still recreational, for some of them, this activity has become addictive with many negative consequences [4,5]. Addiction to video games has been recently recognized as a disorder by current diagnostic systems. More precisely, it appears under the term “internet gaming disorder” (IGD) in Section 3 of the non-substance addiction category of the Diagnostic and Statistical Manual of Mental Disorders, fifth edition (DSM-5) as a condition warranting further research [6], and more recently was included in the 11th International Classification of Diseases (ICD-11) by the World Health Organization (WHO) under the term “gaming disorder” [7,8]. IGD is a major public health issue with a prevalence that varies among countries between 0 and 10 percent [8]. The recognition of IGD as an addictive disorder began in light of several symptoms in common with substance use disorders (SUDs), as well as the observation of similar neural and biological mechanisms [9,10,11,12,13,14]. In the etiological theories of SUDs, personality traits occupy an important factor [15]. Thus, the expression of SUDs would be the association of certain personality traits with environmental factors [16]. It will be the same for IGD, whose emergence would be the conjunction of environmental factors and particular psychological aspects, such as emotion regulation [17] including alexithymia [1], impulsivity [18], sensation seeking [19] and aggressiveness–hostility [20].

Alexithymia is a multidimensional personality trait characterized by disturbances in affective and cognitive function. It is manifested as an inability to find words to describe feelings or emotions [21,22]. Individuals with addictive disorders have shown higher levels of alexithymia compared to control groups [23,24], and alexithymia has long been recognized as a common personality trait in SUDs [25] and a predisposing condition to SUDs [23,26,27]. Individuals diagnosed with IGD have also shown higher levels of alexithymia [1], although it is unclear whether alexithymia in this group is a predisposing trait or a consequence of regular gaming [17]. To assess alexithymia, the Toronto Alexithymia Scale (TAS-20) is commonly used [28], with a significantly higher score in the addicted population compared to healthy controls [23,24]. Impulsivity and sensation seeking are also predictive of addictive behavior [15], and IGD, as well as SUD, have been associated with impulsivity [29,30] and sensation seeking [31]. Impulsivity is an individual’s tendency to behave with little or no regards for consequences [30], or a “lack of reflection” between an environmental stimulus and a behavioral response [32]. In several studies, impulsivity has been stably associated with the early use of substances [33,34] and also to IGD [30,35]. Impulsivity is largely assessed by the Barratt Impulsiveness Scale (BIS-10) in various addicted samples [30]. Sensation seeking is characterized by a greater demand for novelty and exciting experiences and by thrill seeking regardless of associated risks [36]. Usually, sensation seeking is measured by the Sensation Seeking Scale (SSS-V) developed by Zuckerman [37]. There is a strong and long-recognized relationship between sensation seeking and SUD [38]. The relationship between sensation seeking and IGD, particularly in terms of IGD severity, is still debated. Studies conducted so far have concluded either a positive correlation or no relationship [19,39]. Aggressiveness/hostility is often associated with SUD. For specific populations of heroin-dependent individuals, this link is particularly established in the context of domestic violence [40], and a decreased in crime has been reported with improved social functioning as an indicator of the therapeutic efficacy of MMT and of other opiate treatments [41]. In the case of gaming, there is a growing body of evidence invoking the impact of violent video games on aggressive behavior [42,43], and a link between aggressiveness and IGD has been reported in teenagers [44] and young adults populations [45]. However, some authors have suggested that the type of violent video game could induce aggressive or hostile behavior independent of the existence of IGD. Nevertheless, being aggressive is considered a behavior resulting from the interaction between a personality trait and the environment. Thus, violent video games are thought to be environments conducive to aggressive behavior expression [20]. All the studies exploring aggressiveness and hostility dimensions employed heterogeneous clinical parameters from one to another, ranging from the hot sauce paradigm [20] to the objectification of domestic violence [46]. In this context, the Buss-Durkee Hostility Inventory (BDHI), a validated self-administered questionnaire, appears to be a simple way to assess different types of hostility [47].

These personality traits are frequently described in both the SUD and the IGD. However, no study has made a comparison between the two addictive disorders. In this context, it is interesting to assess whether the personality traits described above are found in the same proportions in SUD and IGD. To do so, we selected a sub-population of individuals with IGD from the massively multiplayer online role-playing game (MMORPG) community and a specific subset of SUD population composed of former heroin users currently on MMT. We chose the MMORPG population because it is the most studied population of gamers and is well known to show signs of problematic use more often [1,4,48]. Although IGD affects gamers across many game genres, MMORPGs are particularly popular and thought more likely to be addictive [49]. World of Warcraft (WoW) was the most popular MMORPG worldwide upon its release, gathering a player community of 11 million players in 2008 [50], and it is still one of the most popular fourteen years later [51].

The MMT group was chosen as a representative group for SUD because heroin addiction has been described in the literature as the gold standard in terms of dependence and risk factors [52]. We used valid psychometric scales focusing on the personality traits listed above to compare the IGD and the SUD (MMT) groups in an effort to confirm the presence of similarities that are implied in the previous literature.

## 2. Materials and Methods

### 2.1. Participants

Participants were recruited according to their allocation group. The study was conducted in France and the participants were French. For both groups, participants were asked to certify that they were over 18 of age and to provide informed consent before accessing the questionnaires.

MMORPG gamers with IGD were selected from a French cohort recruited in our online study called Add MMORe, and constituted the MMORPG-IGD group. A cutoff ≥ 3 on the 7 items of the DSM-IV-TR substance dependence adapted scale (DAS) [48] was applied to identify individuals with IGT in this MMORPG sample. Add MMORe was an anonymous online study that recruited French MMORPG not seeking treatment from MMORPGs’ guild forums (private forums of gamers’ groups called guilds in the WoW game where gamers interact and discuss gaming rules and tactics), between May 2009 and March 2010. All subjects gave their consent through electronic signature before starting the questionnaire. More information on the study design and descriptive results (socio-demographic data, internet addiction and gaming disorder screening tools and excessive gaming consequences) have been published previously [48].

Individuals in the MMT group were recruited in an outpatient drug treatment center in Besançon, France. They had to be stabilized under MMT and to had not used heroin for at least 4 weeks at the time of the assessment when recruited. The diagnosis of substance dependence was confirmed according to the DSM-IV-TR criteria in a semi-structured interview, and the SUD was categorized as severe according to the impacts on income in all cases.

The study protocol was approved by the Ethics Committee of Besançon University Hospital (authorization given by the General Health Administration: DGS 2007-0382) [48].

### 2.2. Psychological Assessment Material

#### 2.2.1. Dysfunctional Engagement in MMORPGs, DSM IV-TR for Substance Dependence Adapted Scale (DAS)

The DAS used in this study was published in a previous study as a good discriminative screening tool for MMORPGs addiction [48]. It consisted of seven items (adapted from substance dependence DSM IV-TR criteria) related to online game MMORPG use over a past period of 12 months to which were answered « yes » or « no ». A threshold of ≥3 positive criteria on the 7 items was applied to determine whether MMORPG players exhibit IGD. In fact, the study was conducted (2009–2010), IGD has not yet been introduced into the DSM-5 (2013) and the DSM-IV-TR criterion for addiction was the standard for assessing addictive disorders.

#### 2.2.2. The Toronto Alexithymia Scale (TAS-20)

The Toronto Alexithymia Scale (TAS-20) (score range: 20–100; Cronbach’s α = 0.77) is a 20-item inventory, validated in France, measuring alexithymia [21,28]. A total score ≥ 56 is considered to be within the range of alexithymia. Alexithymia is a personality trait composed of different dimensions, and the TAS-20 examines three sub-components: difficulty identifying feelings (DIF) (score range: 7–35); difficulty describing feelings (DDF) (score range: 5–25); and externally oriented thinking (EOT) (score range: 8–40) [21]. Overall test-retest reliability was assessed over eight months (r = 0.78) [28,53].

#### 2.2.3. The Barratt Impulsiveness Scale, Version 10 (BIS-10)

The Barratt Impulsiveness Scale, version 10 (BIS-10) is a 34-item questionnaire [54,55], validated in France, assessing impulsivity with a global score (score range: 0–136; Cronbach’s α = 0.82) also having three sub-components: attentional impulsivity (AI) (score range: 0–44; Cronbach’s α = 0.56); motor impulsivity (MI) (score range: 0–44; Cronbach’s α = 0.79); and non-planning impulsivity (NPI) (score range: 0–48; Cronbach’s α = 0.64). Each item is scored using 4-point ratings (1 = never/rarely, 2 = occasionally, 3 = often, 4 = almost always/always) to determine impulsivity levels.

#### 2.2.4. The Zuckerman Sensation Seeking Scale, Form V (SSS-V)

The impulsivity trait in the present study has also been measured by the French-validated Zuckerman Sensation Seeking Scale [56], form V (SSS-V) [37]. This 40-item scale assesses compulsion to experience novel and risky experiences according four subscales: thrill and adventure seeking (TAS) (for each item Cronbach’s α ranged from 0.45 to 0.71); experience seeking (ES) (for each item Cronbach’s α ranged from 0.30 to 0.54); disinhibition (DIS) (for each item Cronbach’s α ranged from 0.39 to 0.71); and boredom susceptibility (BS) (for each item Cronbach’s α ranged from 0.34 to 0.45). The total score range was from 0 to 40, and the score for each of the subscales ranged from 0 to 10. High scores corresponded to high levels of impulsivity [56].

#### 2.2.5. Buss-Durkee Hostility Inventory (BDHI)

Aggressiveness (lifetime) was rated with the Buss-Durkee Hostility Inventory [47,57], a 75-item questionnaire including seven subscales measuring hostility traits and guilt: assault (10 items); indirect hostility (9 items); irritability (11 items); negativism (5 items); resentment (8 items); suspicion (10 items); verbal hostility (13 items); and guilt (9 items). Each item has a statement to which the participant responds “true” or “false”. One point is awarded for each item (1 point for true except for some items where the point is assigned for false). We considered “overall” (score range: 0–75; Cronbach’s α = 0.88), behavioral hostility (Cronbach’s α =0.80) which grouped “assault”, “verbal aggression” and “indirect hostility”, cognitive hostility (Cronbach’s α = 0.78) which grouped “resentment” and “suspicion”, and subscales irritability (Cronbach’s α = 0.70) and negativism (Cronbach’s α = 0.53). Overall the test-retest reliability was of three-month (r = 0.87). The higher the scores, the more aggressive the subjects were considered to be [58].

### 2.3. Procedure

Individuals in the MMORPG-IGD group were recruited from discussion forum guilds and those in the MMT group were outpatients of a drug treatment center in Besançon, France. SUD was confirmed in the individuals in the MMT group by semi-structured psychiatric interviews and IGD was defined by the online DAS results.

Despite this difference, all volunteers consented to participate in the online study and the assessment lasted 45 min with the DAS to index problematic online video game behavior and the DSM-IV-TR substance dependence scale. The psychometric scales were assessed in the same following sequence: TAS-20, BIS-10, SSS-V and BDHI. Sociodemographic data were also collected using a questionnaire including gender, age, marital status, employment status and years of education.

### 2.4. Statistical Analysis

Statistical analyses were performed using R version 4.1.0 (R Core Team 2020. R Foundation for Statistical Computing, Vienna, Austria). The characteristics of each group were described as number and percentage for the qualitative variables and as mean and standard deviation for the quantitative variables. The sociodemographic data were compared by Pearson’s chi-squared test (unmatched categorical variables) except for education level, where a Fisher’s exact test was applied. The two groups were compared by bivariate, followed by multivariate, analysis using a multiple linear regression model. In the bivariate analysis, the values of the TAS-20, BIS, SSS-V and BDHI scales were compared between the two groups by using Welch’s tests, an adapted version of the *t*-test for cases of unequal variance. The variances of each scale and subscale were compared between the two groups by Levene’s test. In the multivariate analysis, multiple linear regression was used with adjustment for age and sex. The significance level was set to 0.05.

## 3. Results

### 3.1. Demographics and Clinical Questionnaires

Among the Add MMORe study gamers, 83 met the criteria for IGD and constituted the MMORPG-IGD group [35%, mean age: 25.22 (SD 6.20)]. Forty-seven French outpatients were recruited in the MMT group [mean age: 30.60 (SD 6.92)]. Participant data are reported in Table 1. All of the MMT group members were selected in the drug treatment center and all exhibited a high level of addiction, with all DSM-IV-TR criteria positive and a dependence syndrome requiring MMT. MMORPG-IGD was present in gaming participants who met three or more of the seven online DAS criteria. Not all the individuals in the MMORPG-IGD group exhibited the same levels of addiction (Table 2).

### 3.2. Psychometric Evaluations

Compared to the MMT group, the MMORPG-IGD group presented lower scores on the DDF and DIF components of the TAS-20. The MMORPG-IGD group also had lower scores than the MMT group on the BIS-10 and all of its subscales (IM, IC and NPI); on the SSS-V, only for the ES and DS subscales; and on the total BDHI score, but only on the assault and indirect hostility subscales (Table 3).

There were significant differences between the two groups in the scores on the TAS-20 (*p* = 0.03) and its DDF subcomponent (*p* = 0.006), and on the SSS-V (*p* < 0.001) and its ES and DS subcomponents (*p* = 0.016, ES, and <0.001, DS) (see Table 2 and Figure 1).

These score differences remained significant in the multivariate analysis (Table 4). R-squared values for each model are presented in Table 5.

The radar chart in Figure 2 shows the data from the different psychometric scales as percentages, and shows that the IGD and SUD groups have similar and relatively transposable personality profiles, differing only in intensity.

## 4. Discussion

To investigate whether gamers qualifying for IGD share personality traits with other individuals suffering from other kinds of addictive disorders, we focused on the personality traits described in the literature as being overexpressed in individuals suffering from SUD. Overall, for both populations, a similar, relatively transposable (Figure 2) personality profile emerged, with higher scores for the MMT group than for the MMORPG-IGD group. The lower scores in the MMORG-IGD group may have been a function of the greater variance in the severity of addiction in that group.

People with addictive disorders have been already characterized as more prone to alexithymia than controls, even when evaluated after a prolonged period of abstinence [59]. However, no distinction between different addictive disorders had been made. Here, we observed significantly higher levels of alexithymia in the MMT population than in the MMORPG-IGD group, at TAS-20 total scores, TAS-20 DIF and DDF sub-scores. However, no statistical difference was found between groups for the EOT sub-score. On one hand, alexithymia has been described as a stable personality trait that constitutes an important underlying risk factor for SUD [60] or for behavioral addictions [61] including IGD [1]. On the other hand, some authors assert that alexithymia is not a stable personality trait, noting that the DIF and DDF sub-dimensions of the TAS-20 score can be influenced by mood or anxiety at the time of the evaluation [60,62], whereas the EOT dimension remains stable [1] with no emotional impact [63]. The influence of methadone on the ability to describe and to identify feelings must also be considered, since methadone treatment is well known for inducing blunted emotional reactivity in both elative and depressive states [64]. The greater variance in DDF also suggested that, as noted, the MMORPG-IGF group was more heterogeneous, and that this heterogeneity might be explained by the high variance in the severity of addiction quantified by the number of DAS criteria in this group (see Table 2). To conclude, EOT appeared to be a stable dimension among individuals in both of our study groups, whereas the other sub-dimensions of the TAS-20 may be unstable and more greatly influenced by factors such as treatments or addiction severity.

Impulsivity has long been described as common to all addictions and as a central trait of an addictive personality [65]. We measured impulsiveness by the BIS-10. The MMT group had a higher score than the MMORPG-IGD group. Gamers with greater impulsivity have previously been highlighted as being at higher risk for IGD [66]. Similarly, greater impulsivity has been implicated in heroin dependence [59]. The relationship between impulsivity and different addiction types appears to be more complex than just a class effect [35]. Thus, if impulsivity levels are described as independent from the substance type [67], consumption of the substance itself seems to influence behavior and use patterns [68]. As a rule, heroin use, as compared to internet gaming, is an illicit activity that seems to require greater impulsivity traits and risk-taking levels [69]. Moreover, it is important to consider the complex link between cognitive issues and video gaming. Apart from addiction, gamers may have enhanced attentional skill due to the training aspect of gaming for discriminating relevant and irrelevant visual stimuli [70], along with improved cognitive control [71], and IGD severity has been described as negatively correlated to impulsivity [72]. This has been attributed to the strategic nature of the MMORPG attracting a population with lower impulsivity. Indeed, a high level of impulsivity confers such disadvantage in an MMORPG that it inevitably leads to displeasure in practice and therefore protects from IGD [72]. Thus, while MMORPG-IGD may, like SUD, exhibit greater impulsivity, it is not at the same intensity, and might be attenuated by the cognitive training required by the game and the development of functional impulsivity.

Sensation seeking has previously been argued to be a personality trait common among individuals with SUD [38], and we observed a higher score on the SSS-V in MMT than in MMORPG-IGD, particularly in the ES and DIS subscales. The DIS aspect encompasses extroversion, social disinhibition, a variety of sexual desires and the use of psychoactive substances and it has previously been associated with SUD, and more particularly with consumption initiation [73]. ES reflects the search for novelty and the tendency to adopt an unconventional lifestyle, while BS reflects an aversion to routine activities. Thus, the fact that the MMORPG-IGD group had a lower DIS score appears to be relevant and in agreement with the literature. MMORPG players have been described as shy and with feelings of social insecurity [31], and with regard to ES behaviors, it is known that IGD is associated with lower levels of extroversion and less openness to new experiences [39]. In the remaining aspect of the SSS-V, the MMORPG-IGD group showed the same level of TAS—which is translated in the literature as the attraction to thrills, such as risky sports—as the MMT group.

These results drive us to the hypothesis that in a virtual universe, participants are able to manage the different aspects of sensation seeking by satisfying each of them. Perhaps the virtual universe will prove to be a good compromise to satisfy TAS and resist BS without exposure to a social or new environment that individuals with IGD cannot control. Sensation seeking provides a coping mechanism to overcome boredom [19]. While we found that the MMT group was more impulsive and had higher scores in the SSS-V than the MMORPG-IGD group, we also observed a higher variance in MMORPG-IGD than in MMT on the SSS-V and its ES and DIS subscales, and, as previously, we cannot exclude the influence of differences in addiction severity on these aspects. Indeed, our MMORPG-IGD group was more heterogeneous in terms of addiction severity with a range of positive DAS criteria between three and seven. However, the number of individuals in the MMORPG-IGD group with seven positive criteria was not sufficient (*n* = 3) to perform statistics and answer whether it is the severity more than the object of the addiction that causes greater impulsivity.

Finally, by using the BDHI, we compared aggressiveness in MMT and IGD, both of which have previously been associated with an increased propensity for aggressive behavior. The MMT group had a higher overall aggressiveness score on the BDHI than the MMORPG-IGD group. This scale assesses different aspects of hostility with two general factors: neurotic hostility, specifically resentment, suspicion and guilt; and anger expression, specifically assault, indirect hostility, verbal hostility, irritability and negativism [74,75]. The analysis of each of these specific subscales showed that the only significant differences between groups were the expressive hostility categories of assault and indirect hostility. This is not surprising because these anger expression subscales are defined respectively as physical violence against others and roundabout and indirect aggressive behavior [76]. On the other hand, if gamers have a reputation of being aggressive, it is important to emphasize that this aggression is not expressed in general at the same level as in SUD. Exposure to violent gaming seems to induce emotional desensitization to actual violence [77] and lead to an increase in aggressive behavior [78,79], particularly in IGD [80]. Nevertheless, so far these findings have described only virtual behaviors [81], and it is important to note that no study has yet reported aggressiveness in IGD as having the same legal implications, i.e., domestic violence and criminal activities, as the aggressiveness reported in some cases of heroin dependence [40,82,83].

Concerning the overall personality profile, and in particular the comparison as allowed by the radar plot in Figure 2, we confirmed that personality traits in MMT and IGD are globally similar and even stackable. Our main observations point in the direction of a higher intensity for the MMT group, which could be associated with addiction severity and/or environmental factors. Moreover, at least for SUD, these whole personality traits are known to co-exist [24]. More precisely, impulsivity and hostility might be a mode of regulating emotion in alexithymic individuals with addiction [59], and the lesser intensity observed in the MMORPG-IGD group may have originated in its variance. Firstly, regarding the participant data, the MMORPG-IGD group was younger than the MMT group, and this age difference should be taken into consideration because we believe it confirms a different time course between the two groups. The entire MMT group had dependence syndrome, conferring homogeneity in addiction severity, whereas the MMORPG-IGD group had varying levels of addiction severity. The MMT group were outpatients recruited from a drug treatment center, whereas we were not aware of any treatment for IGD in the MMORPG-IGD group. These aspects indicate a different stage in the addictive trajectory between the two groups and a different level of addictive disorder apprehension. The MMORPG-IGD group may not have reached the same stage of change pattern as the MMT group, as described by the model of Prochaska and DiClemente [84]. Thus, to determine whether the intensity of the personality traits is attributable to the type of addiction or its severity, the analysis must be continued with groups matched according to the severity of their addiction.

This study has several limitations, a number of which regarding MMORPG populations have been previously described [48]. Our results are also limited by the use of multiple self-reports. For example, we focused only on impulsivity as assessed by the BIS-10, whereas in these populations different types of impulsivity can also be described in terms of functional and dysfunctional impulsivity [72,85]. Finally, several of our psychometric tests and classifications have been updated since our study was conducted, as is the case with the BIS-11 and the DSM-5. We believe that despite this, our results remain relevant and provide further support for the classification of IGD as an addictive behavior. Indeed, at the time of data collection (between 2009 and 2010), many of these self-reports were not yet updated. Given the many changes that have occurred in the video game industry over the past decade [86], we might be concerned about the representativeness of our sample. However, we have observed that this change in gaming had already taken place before 2010 [87,88]. Moreover, the MMORPG universe remains stable, WoW is still popular [51] and the practice remains overrepresented by men and young people [49,89]. However, all these data were obtained before the 2019 coronavirus pandemic (COVID-19), which is known to have increased the prevalence of IGD [90] due to its very particular social timing. This simply reminds us that addiction emergence is not the simple expression of an individual vulnerability, but the conjunction of different phenomena leading to the biopsychosocial model [91]. In view of these different elements, the data collected on the personality traits of the MMORPG-IGD group still seem to be representative and relevant.

## 5. Conclusions

In conclusion, we observed common personality traits between the MMORPG-IGD and MMT-SUD groups that might be involved in vulnerability to addiction. The expression of these personality traits may influence the choice of one behavior over another, or even the severity of the addiction, and in turn, the intensity of these traits may be affected by some direct environmental factors, i.e., pharmacological treatment or habit exposure. The objectification of similar personality traits between these two very different populations brings provide new evidence to support the conceptualization of IGD as an addiction like any other. Thus, certain personality traits, such as shyness and social insecurity, could influence the orientation towards MMORPG rather than heroin addiction. The virtual universe offers an excellent compromise to satisfy certain impulsive and sensation-seeking traits with a universe full of stimulation while limiting exposure to a social network. These results provide important information for better understanding how specific personality traits can interact with the environment in the emergence of an addiction and direct an individual towards a specific addiction (here, heroin or IGD). Moreover, they provide an understanding of the personality functioning of each of the addictive groups, which allows us to better apprehend them in build care projects. Finally, this understanding will allow us to better design specific and common prevention actions for each group. Investigations should be continued with the MMORPG-IGDs group only, which has the highest addiction scores in order to determine whether the higher intensity of personality traits observed in the MMT group is better explained by the severity of the addiction or by environmental factors.

## Figures and Tables

**Figure 1 ijerph-19-09536-f001:**
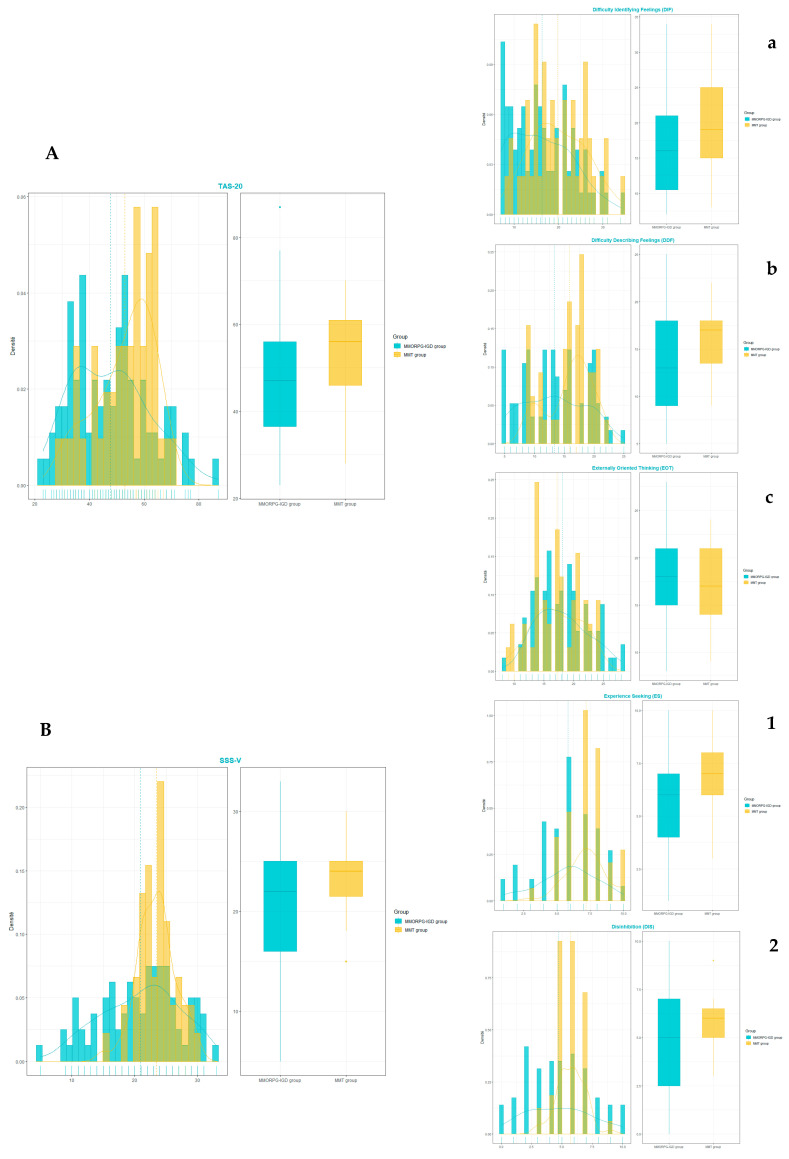
Histogram and box plot comparing the score distributions between MMORPG-IGD and MMT. (**A**) on the TAS-20 and its subcomponents, (**a**) Difficulty Identifying Feelings (DIF), (**b**) Difficulty Describing Feelings (DDF) and (**c**) Externally Oriented Thinking (EOT); (**B**) on the SSS-V and its subcomponents, (**1**) Experience Seeking (ES) and (**2**) Disinhibition (DIS). MMORPG-IGD, massively multiplayer online role-playing game internet gaming disorder; MMT, methadone maintenance therapy; TAS-20, Toronto Alexithymia Scale; SSS-V, Zuckerman Sensation Seeking Scale, form V.

**Figure 2 ijerph-19-09536-f002:**
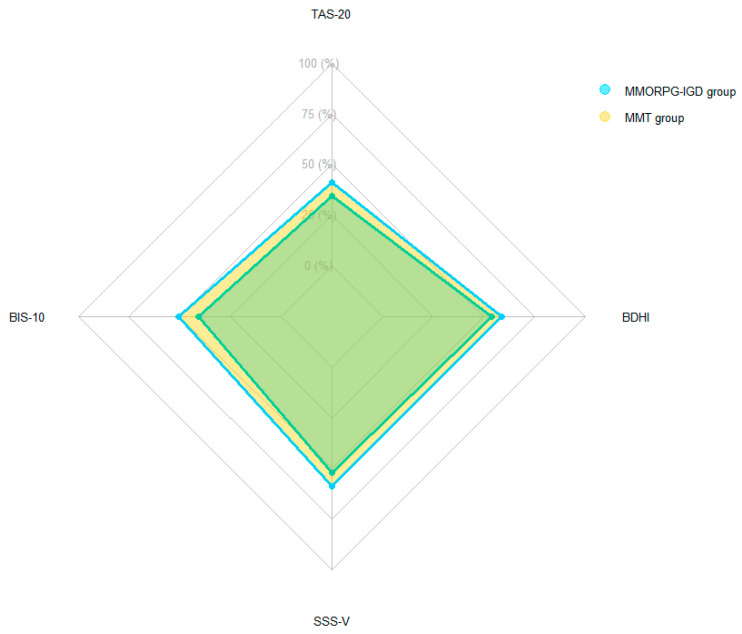
Radar plot of psychometric scales for both groups. MMORPG-IGD, massively multiplayer online role-playing game internet gaming disorder; MMT, methadone maintenance therapy; TAS-20, Toronto Alexithymia Scale; BIS-10, Barratt-Impulsiveness Scale, version 10; BDHI, Buss-Durkee Hostility Inventory; SSS-V, Zuckerman Sensation Seeking Scale, form V.

**Table 1 ijerph-19-09536-t001:** Participant data.

	Total Sample (*n* = 130) ^1^	MMORPG-IGD Group (*n* = 83) ^1^	MMT Group (*n* = 47) ^1^	*p*-Value ^2^
**Age**	27.16 (6.95)	25.22 (6.20)	30.60 (6.92)	<0.001
**Gender male: *n* (%)**	103 (79.2%)	71 (85.5%)	32 (68.1%)	0.018
**Non-single: *n* (%)**	43 (33.1%)	28 (33.7%)	15 (31.9%)	0.832
**With children: *n* (%)**	27 (21.1%)	15 (18.3%)	12 (26.1%)	0.300
**Domicile**				
Lives with someone else (parents, couple, roommate)	41 (31.5%)	19 (22.9%)	22 (46.8%)	0.005
Parental home	34 (26.2%)	29 (34.9%)	5 (10.6%)	0.002
**Education Level**				
Less than high school	29 (22.7%)	7 (8.4%)	22 (48.9)	<0.001
High school graduates	87 (68.0%)	64 (77.1%)	23 (51.1%)	
Some college or more	12 (9.4%)	12 (14.5%)	0 (0.0%)	
**Employment status**				
Unemployed	28 (22.8%)	2 (2.6%)	26 (55.3%)	<0.001
Student	35 (28.5%)	30 (39.5%)	5 (10.6%)	
Employed	60 (48.8%)	44 (57.9%)	16 (34.0%)	

^1^ Mean (SD), *n* (%). ^2^ Two Sample *t*-test; Pearson’s chi-squared test; Fisher’s exact test. MPORG-IGD, massively multiplayer online role-playing game internet gaming disorder; MMT, methadone maintenance treatment.

**Table 2 ijerph-19-09536-t002:** Number of DAS criteria met in the MMORPG-IGD group.

Number of Positive DAS Criteria	MMORPG-IGD (83) ^1^
3	38 (45.8%)
4	24 (28.9%)
5	12 (14.5%)
6	6 (7.2%)
7	3 (3.6%)

^1^*n* (%). DAS, DSM-IV-TR substance dependence adapted scale; MMORPG-IGD, massively multiplayer online role-playing game internet gaming disorder.

**Table 3 ijerph-19-09536-t003:** Comparison of psychometric scores.

	MMORPG-IGD (83) Mean (SD)	MMT (47) Mean (SD)	*p*-Value
**TAS-20**	**47.71 (14.11)**	**52.91 (10.75)**	**0.020**
Difficulty Describing Feelings (DDF)	13.34 (5.30)	15.85 (3.82)	0.002
Difficulty Identifying Feelings (DIF)	16.29 (6.83)	19.87 (6.33)	0.003
Externally Oriented Thinking (EOT)	18.08 (4.43)	17.19 (4.06)	0.247
**BIS-10**	**55.14 (17.37)**	**68.51 (14.54)**	**<0.001**
Motor Impulsivity (MI)	16.81 (7.20)	22.26 (6.29)	<0.001
Attentional Impulsivity (AI)	18.00 (6.82)	21.36 (6.30)	0.005
Non-Planning Impulsivity (NPI)	20.34 (7.42)	24.89 (6.95)	<0.001
**SSS-V**	**20.89 (6.26)**	**23.45 (3.06)**	**0.002**
Thrill and Adventure Seeking (TAS)	6.49 (2.03)	6.74 (1.70)	0.454
Experience Seeking (ES)	5.82 (2.15)	7.19 (1.48)	<0.001
Disinhibition (DIS)	4.71 (2.67)	5.72 (1.16)	0.003
Boredom Susceptibility (BS)	3.87 (1.99)	3.79 (1.44)	0.792
**BDHI**	**40.57 (9.89)**	**44.19 (9.42)**	**0.041**
** *Hostility traits* **	35.41 (9.42)	38.32 (9.17)	0.088
Assault	4.23 (3.00)	5.49 (2.87)	0.020
Indirect hostility	4.65 (2.01)	5.55 (1.72)	0.008
Irritability	6.45 (1.92)	6.96 (2.23)	0.190
Negativism	2.84 (1.43)	2.94 (1.29)	0.706
Resentment	4.45 (2.28)	4.17 (2.17)	0.496
Suspicion	4.99 (2.52)	5.45 (1.98)	0.253
Verbal hostility	7.81 (2.40)	7.77 (2.11)	0.919
** *Guilt* **	5.16 (2.17)	5.87 (2.38)	0.093

TAS-20, Toronto Alexithymia Scale; BIS-10, Barratt Impulsiveness Scale, version 10; SSS-V, Zuckerman Sensation Seeking Scale, form V; BDHI, Buss-Durkee Hostility Inventory.

**Table 4 ijerph-19-09536-t004:** Results of multivariate linear regression model analysis, adjusted for age and gender.

	Crude Estimates (SD)	*p*-Value	Adjusted Estimates (SD)	*p*-Value
TAS-20	5.2 (2.37)	0.030	6.91 (2.62)	0.009
BIS-10	13.37 (3)	0.000	16.8 (3.25)	0.000
SSS-V	2.56 (0.97)	0.010	2.71 (1.09)	0.014
BDHI	3.63 (1.78)	0.043	5.51 (1.92)	0.005

**Table 5 ijerph-19-09536-t005:** Adjusted R-squared values for multiple linear regression.

Dependent Variable	Adjusted R^2^
TAS-20	0.032
BIS-10	0.162
SSS-V	0.030
BDHI	0.071

## Data Availability

All data can be provided by our methodologist Frédéric Mauny upon request by e-mail; frederic.mauny@univ-fcomte.fr.

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
