# Peer review of "Do Individuals with Internet Gaming Disorder Share Personality Traits with Substance-Dependent Individuals?"

_ijerph, 2022, doi:10.3390/ijerph19159536_

Round 1

Reviewer 1 Report

I would like to thank the authors and editors for the opportunity to read this manuscript. The authors raised a very important and interesting topic.

Even before the pandemic, Internet gaming addiction (IGA) seemed to be a public health problem. With the increasing digitization of life and the isolation caused by COVID-19, the importance of this issue is growing.

In the introduction, the authors rightly pointed to the similarities between IGA and substance use disorder (SUD). The results of the research conducted so far on the relationship of personality traits with both SUD and IGA are also presented. The authors indicated a gap in the current state of knowledge that is filled by their research.

I have a few specific comments:

·        When was the data collected? The authors point to an article from 2011, was the data collected for this work then? If so, please indicate in the limitations whether the obtained results remain up-to-date in the light of the passage of time, progressing digitization and the COVID-19 pandemic.

·        It is worthwhile to state the Cronbach alpha coefficients of the tools used.

·        The Barratt-Impulsiveness Scale, The Zuckerman Sensation Seeking Scale, Buss-Durkee Hostility Inventory - it is worth adding, how to interpret the obtained result.

·        The authors admit: “While we found that the MMT group was more impulsive and had higher scores in the SSS-V than the MMORPG-IGD group, we also observed a higher variance in MMORPG-IGD than in MMT on the SSS-V and its ES and DIS sub-scales, and as previously, we cannot exclude the influence of differences in addiction severity on these aspects” – did the authors consider including the only participants with higher levels of IGD in the study?

Author Response

REVIEWER 1

  1. When was the data collected? The authors point to an article from 2011, was the data collected for this work then? If so, please indicate in the limitations whether the obtained results remain up-to-date in the light of the passage of time, progressing digitization and the COVID-19 pandemic.

We are grateful to the reviewer for pointing this out. We have added the inclusion period in the method. In addition, due to some confusion induced by our presentation, we have made some changes at the request of your last comment and another reviewer's comment. Thus, in the method section we provide more details on the recruitment of our population and in the result, we give the number of subjects retained for analysis.

Line 564-575 Methodological section

“MMORPG gamers with IGD, were selected from a French cohort recruited in our online study called Add MMORe, and constituted the MMORPG-IGD group. A cutoff ≥3 on the 7 items of the DSM-IV-TR substance dependence adapted scale (DAS) [48] was applied to determine in this MMORPG sample those qualifying for IGD. Add MMORe is an anonymous online study which recruited French MMORPG non-treatment seeking gamers from MMORPGs’ guild forums (private forums of gamers’ groups called guilds in WoW game where gamer's interact and discuss about gaming rules and tactic), between May 2009 and March 2010. All subjects gave their consent through electronic signature before starting the questionnaire. More information about the study design and descriptive results (socio demographic data, Internet addiction and gaming disorder screening tools and excessive gaming consequences) have previously been published [48].”

Line 924-926 Result section

“Among the Add MMORe study gamers, 83 met the criteria for IGD and constituted the MMORPG-IGD group [35%, mean age: 25.22 (SD 6.20)]. Forty-seven French outpatients were recruited in the MMT group [mean age: 30.60 (SD 6.92)].”

In addition, we agree with the comment about the evolution of the video game industry over the last decade. But we are not likely to be concerned about the representativeness of our sample according to some studies which have identified that this evolution of gaming did not impact MMORPGs. Indeed, MMORPGs remain stable in their game structure and they remain popular. In particular, WoW is an MMORPG that continues to gather the second largest MMORPG community. Furthermore, the socio-demographic data collected in 2010 remains comparable to more recent data, suggesting that our sample remains representative. We have added this information in the discussion section and note that we discuss the COVID pandemic

Line 1209-1233

“Indeed, at the time of data collection (between 2009 and 2010), many of these self-reports were not yet updated. In front of the many changes in the video game industry this last decade [86], we could be worry about the representatively of our sample. However, we observed that this gaming change already occurred before 2010 [87,88]. In addition, the MMORPG universe remains stable, WoW popular [51] and practice stays overrepresent by men and young people [49,89]. However, all these data were obtained before the 2019 coronavirus pandemic (COVID-19) which is known to have increased the prevalence of IGD [90], due to its very particular social moment. This aspect simply reminds us that the addiction emergence is not the simple expression of an individual vulnerability but the conjunction of different phenomena leading to the biopsychosocial model [91]. In view of these different elements, the data collected on the personality traits of the MMORPG-IGD group still seems to be representative and relevant. “

  1. It is worthwhile to state the Cronbach alpha coefficients of the tools used.

                  We are sorry, we did not think that it was necessary to present the Cronbach alpha because these tools were validated many years ago, and these resultats published. We correct that by adding the Cronbach alpha available in literature for each psychometric test.

  1. The Barratt-Impulsiveness Scale, The Zuckerman Sensation Seeking Scale, Buss-Durkee Hostility Inventory - it is worth adding, how to interpret the obtained result.

                  We agree with your comment and we added some details on the interpretation, from the line 584 to 841:

“2.2.1. Dysfunctional engagement in MMORPGs, DSM IV-TR for substance dependence Adapted Scale (DAS)

DAS used in this study has been published previously as a good discriminating screening tool for addiction to MMORPGs [48]. It consisted In seven items (adapted from substance dependence DSM IV-TR criteria) related to online game MMORPG use over a past period of 12-months, answered « yes » or « no ». A threshold of ≥3 positive criteria on the 7 items was applied to determine whether MMORPG players qualified for IGD. In fact, at the time the study took place (2009-2010), IGD was not yet includced in the DSM and the DSM-IV-TR criterion for addiction was the standard for assessing addictive disorders.

2.2.2. The Toronto Alexithymia Scale (TAS-20)

The Toronto Alexithymia Scale (TAS-20) (score range: 20-100; Cronbach’s α = 0.77) is a French-validated 20-item inventory measuring alexithymia [21,28]. A total score ≥56 is considered as being within the alexithymia range. Alexithymia is a personality trait composed of different dimensions, and the TAS-20 examines three sub-components: difficulty identifying feelings (DIF) (score range: 7-35); difficulty describing feelings (DDF) (score range: 5-25); and externally oriented thinking (EOT) (score range: 8-40)[21]. The overall have an eight months test-retest reliability (r = 0.78) [28,53]

2.2.3. The Barratt-Impulsiveness Scale, version 10 (BIS-10)

The Barratt-Impulsiveness Scale, version 10 (BIS-10) is a French-validated 34-item questionnaire [54,55] assessing impulsivity with a global score (score range: 0-136; Cronbach’s α = 0.82), also with three sub-components: attentional impulsivity (AI) (score range: 0-44; Cronbach’s α = 0.56) ; motor impulsivity (MI) (score range: 0-44; Cronbach’s α = 0.79); and non-planning impulsivity (NPI) (score range: 0-48; Cronbach’s α = 0.64). Each item is scored using 4-point ratings (1=never/rarely, 2=occasionally, 3=often, 4=almost always/always) to determine impulsivity levels.

2.2.4. The Zuckerman Sensation Seeking Scale, form V (SSS-V)

The impulsivity trait in the present study, was additionally measured by the French-validated Zuckerman Sensation Seeking Scale [56], form V (SSS-V) [37]. This 40-item scale evaluates the compulsion to novel and risky experiences along four subscales: thrill and adventure seeking (TAS) (for each item Cronbach’s α the ranged from 0.45 to 0.71); experience seeking (ES) (for each item Cronbach’s α the ranged from 0.30 to 0.54); disinhibition (DIS) (for each items Cronbach’s α the ranged from 0.39 to 0.71); and boredom susceptibility (BS) (for each item Cronbach’s α the ranged from 0.34 to 0.45). The total score ranges from 0 to 40, as well as each of the subscales ranges from 0 to 10. High scores correspond to high levels of impulsivity [56].

2.2.5. Buss-Durkee Hostility Inventory (BDHI)

Aggressiveness (lifetime) was rated with the Buss-Durkee Hostility Inventory [47,57], a 75-item questionnaire including seven subscales measuring hostility traits and guilt: assault (10 items); indirect hostility (9 items); irritability (11 items); negativism (5 items); resentment (8 items); suspicion (10 items); verbal hostility (13 items); and guilt (9 items). Each item has a statement to which the participant answers "true" or "false". One point is awarded for each item (1 point for true except for some items where the point is awarded for false). We considered “overall” (score range: 0-75; Cronbach’s α = 0.88), behavioral hostility (Cronbach’s α =0.80) that grouped “assault”, “verbal aggression” and “indirect hostility”, cognitive hostility (Cronbach’s α =0.78) that grouped “resentment” and “suspicion”, and subscales: irritability (Cronbach’s α =0.70) and negativism (Cronbach’s α =0.53). The overall have a three months test-retest reliability (r = 0.87). The higher the scores, the more aggressive the subjects are considered to be [58].”

  1. The authors admit: “While we found that the MMT group was more impulsive and had higher scores in the SSS-V than the MMORPG-IGD group, we also observed a higher variance in MMORPG-IGD than in MMT on the SSS-V and its ES and DIS sub-scales, and as previously, we cannot exclude the influence of differences in addiction severity on these aspects” – did the authors consider including the only participants with higher levels of IGD in the study?

Thank you to the reviewer for pointing this out. We wanted to discuss this aspect but in view of your comment we realize that our discussion is not sufficient and suffers from a lack of precision. That is why, for more clarity, we have added sentences to better specify our limits. Indeed, our population is too heterogeneous but selecting only the IGD subjects with the highest addiction severity is statistically impossible due to their low proportion as you can see in Table 2.

See Line 1143-1148:

“Indeed, our MMORPG-IGD group was more heterogeneous in terms of addiction severity with a range of positive DAS criteria between 3 and 7. However, the number of individuals in the MMORPG-IGD group with 7 positive criteria was not sufficient (n=3) to perform statistics and answer whether it is the severity more than the object of the addiction that causes greater impulsivity.”

Reviewer 2 Report

The article is well written and the topic is very interesting. But the introduction is weak. I think it is necessary to write it longer and include more references. And the other question is the conclusion. This part needs to be expanded further.

Author Response

REVIEWER 2

  1. The article is well written and the topic is very interesting. But the introduction is weak. I think it is necessary to write it longer and include more references. And the other question is the conclusion. This part needs to be expanded further.

We thank the reviewer for this comment. We have added information and references in the introduction and completed the conclusion for clarity. We hope you will be satisfied with this change.

Reviewer 3 Report

The paper aimed to investigate similarities and differences between individuals affected by IGD and Substance dependence. Although the topic of the study is interesting, the theoretical foundation of the study is not well justified. Therefore, the Authors should clarify the relationships among the constructs and their importance for the study's goal. Finally, a theoretical justification should be included.

Method

For each measure, please provide the values of reliability.

Discussion

Please provide both social and practical implications of the results.

Author Response

REVIEWER 3

  1. The paper aimed to investigate similarities and differences between individuals affected by IGD and Substance dependence. Although the topic of the study is interesting, the theoretical foundation of the study is not well justified. Therefore, the Authors should clarify the relationships among the constructs and their importance for the study's goal. Finally, a theoretical justification should be included.

We are grateful to the rewiever. We added in introduction and in discussion some element to bring more theoretical foundation and justify more the interest of our papers.

  1. Method: For each measure, please provide the values of reliability.

Unfortunately, we did not find reliability values for all measures. Despite this, we added reliability when it was available. We added Cronbach's coefficient to attest to their validity and the internal consistency of each measure.

We agree with your comment and we added some details on the interpretation, from the line 584 to 841:

“2.2.1. Dysfunctional engagement in MMORPGs, DSM IV-TR for substance dependence Adapted Scale (DAS)

DAS used in this study has been published previously as a good discriminating screening tool for addiction to MMORPGs [48]. It consisted In seven items (adapted from substance dependence DSM IV-TR criteria) related to online game MMORPG use over a past period of 12-months, answered « yes » or « no ». A threshold of ≥3 positive criteria on the 7 items was applied to determine whether MMORPG players qualified for IGD. In fact, at the time the study took place (2009-2010), IGD was not yet includced in the DSM and the DSM-IV-TR criterion for addiction was the standard for assessing addictive disorders.

2.2.2. The Toronto Alexithymia Scale (TAS-20)

The Toronto Alexithymia Scale (TAS-20) (score range: 20-100; Cronbach’s α = 0.77) is a French-validated 20-item inventory measuring alexithymia [21,28]. A total score ≥56 is considered as being within the alexithymia range. Alexithymia is a personality trait composed of different dimensions, and the TAS-20 examines three sub-components: difficulty identifying feelings (DIF) (score range: 7-35); difficulty describing feelings (DDF) (score range: 5-25); and externally oriented thinking (EOT) (score range: 8-40)[21]. The overall have an eight months test-retest reliability (r = 0.78) [28,53]

2.2.3. The Barratt-Impulsiveness Scale, version 10 (BIS-10)

The Barratt-Impulsiveness Scale, version 10 (BIS-10) is a French-validated 34-item questionnaire [54,55] assessing impulsivity with a global score (score range: 0-136; Cronbach’s α = 0.82), also with three sub-components: attentional impulsivity (AI) (score range: 0-44; Cronbach’s α = 0.56) ; motor impulsivity (MI) (score range: 0-44; Cronbach’s α = 0.79); and non-planning impulsivity (NPI) (score range: 0-48; Cronbach’s α = 0.64). Each item is scored using 4-point ratings (1=never/rarely, 2=occasionally, 3=often, 4=almost always/always) to determine impulsivity levels.

2.2.4. The Zuckerman Sensation Seeking Scale, form V (SSS-V)

The impulsivity trait in the present study, was additionally measured by the French-validated Zuckerman Sensation Seeking Scale [56], form V (SSS-V) [37]. This 40-item scale evaluates the compulsion to novel and risky experiences along four subscales: thrill and adventure seeking (TAS) (for each item Cronbach’s α the ranged from 0.45 to 0.71); experience seeking (ES) (for each item Cronbach’s α the ranged from 0.30 to 0.54); disinhibition (DIS) (for each items Cronbach’s α the ranged from 0.39 to 0.71); and boredom susceptibility (BS) (for each item Cronbach’s α the ranged from 0.34 to 0.45). The total score ranges from 0 to 40, as well as each of the subscales ranges from 0 to 10. High scores correspond to high levels of impulsivity [56].

2.2.5. Buss-Durkee Hostility Inventory (BDHI)

Aggressiveness (lifetime) was rated with the Buss-Durkee Hostility Inventory [47,57], a 75-item questionnaire including seven subscales measuring hostility traits and guilt: assault (10 items); indirect hostility (9 items); irritability (11 items); negativism (5 items); resentment (8 items); suspicion (10 items); verbal hostility (13 items); and guilt (9 items). Each item has a statement to which the participant answers "true" or "false". One point is awarded for each item (1 point for true except for some items where the point is awarded for false). We considered “overall” (score range: 0-75; Cronbach’s α = 0.88), behavioral hostility (Cronbach’s α =0.80) that grouped “assault”, “verbal aggression” and “indirect hostility”, cognitive hostility (Cronbach’s α =0.78) that grouped “resentment” and “suspicion”, and subscales: irritability (Cronbach’s α =0.70) and negativism (Cronbach’s α =0.53). The overall have a three months test-retest reliability (r = 0.87). The higher the scores, the more aggressive the subjects are considered to be [58].”

  1. Discussion: Please provide both social and practical implications of the results.

As you request, we added many information about the social and practical implication of the result and especially in the conclusion part.

Line 1234-1254:

“To conclude, we observed common personality traits in MMORPG-IGD and MMT-SUD groups that could be involved in vulnerability to addiction. The expression of these personality traits may influence the choice of one behavior over another, or even the severity of the addiction, and in turn, the intensity of these traits may be affected by certain direct environmental factors, i.e. pharmacological treatment or exposure to the habit. The objectification of similar personality traits between these two very different populations brings new elements to support the conceptualization of IGD as an addiction like any other. Thus, certain personality traits such as shyness and social insecurity could influence the orientation towards the MMORPG rather than the heroine. The virtual universe offers an excellent compromise to satisfy certain impulsive and sensation-seeking traits with a universe full of stimulation while limiting exposure to a social network. These results provide important information to better understand how specific personality traits can interfere with the environment in the emergence of an addiction and direct it towards a specific addiction (here heroin or IGD). Moreover, it brings an understanding of the personality functioning of each of the addictive groups which allows to better apprehend them in build care projects. Finally, this understanding will allow us to better design specific and common prevention actions for each group. Investigations should be continued with only the MMORPG-IGDs that have the highest addiction scores to determine if the higher intensity of personality traits observed in the MMT group is better explained by addiction severity or environmental factors.”

Reviewer 4 Report

This is a neat article that attempts to answer the question as to whether individuals with IGD share personality trait with individuals with SUD. There has been a recent interest in investigating whether excessive video game play can/should be characterized as an addiction, and this article further adds to this literature. The article concludes that because individuals with IGD do in fact share personality traits with SUD, that this in turn provides further rationale for considering excessive video game play as an addiction.

The article has several strengths. The methodology section, for the most part, provides a detailed account and overview of the study, highlighting measures that were used. The results section is highly detailed and contains helpful charts and tables. 

My suggestions and recommendations, broadly speaking, pertain to the verbiage of IGD. On lines 35-37, the authors write about IGD being in the DSM-5. I believe there needs to be clarity regarding the fact that IGD is categorized in the DSM-5 as a categorization for further study rather than an actual disorder. At the time of publication in 2013, there was insufficient evidence to suggest that IGD be a disorder (similar to caffeine use disorder). There needs to also be more background in the introduction regarding IGD (what it is, how it has been operationalized and studied) as there are only about 8 lines that describe it. 

In the methodology section, I believe there needs to be more information about the "forum guilds" that were used to recruit participants. For instance, what is a forum guild, and why was a forum guild used? Provide more detail for what MMORPGs are and why they were used in participant recruitment. Regarding the adapted measure for IGD, there needs to be more background regarding how it is used for measurement. In addition, why was this measure used over others given that this is a rather novel construct? Lastly, why were the other psychological assessments utilized over other measurements? Please provide some more rationale for why you chose your measurements.

Author Response

  1. English language and style are fine/minor spell check required

We checked English language and spelling by an extern organism “BioMedProofreading”.

  1. My suggestions and recommendations, broadly speaking, pertain to the verbiage of IGD. On lines 35-37, the authors write about IGD being in the DSM-5. I believe there needs to be clarity regarding the fact that IGD is categorized in the DSM-5 as a categorization for further study rather than an actual disorder. At the time of publication in 2013, there was insufficient evidence to suggest that IGD be a disorder (similar to caffeine use disorder). There needs to also be more background in the introduction regarding IGD (what it is, how it has been operationalized and studied) as there are only about 8 lines that describe it. 

We totally agree with reviewer on this aspect. We decided as request to add some element about the IGD.

Line 36-44

“Thus, if for most of the consumers, this activity still recreational, for another part of them, this activity become addictive with many negative consequences [4,5]. Addiction to video game is recently recognized as a disorder by current diagnostic systems. More precisely, it appears under the term of “Internet Gaming Disorder” (IGD) in the section 3 for the non-substance addiction category of the Diagnostic and Statistical Manuel of Mental Disorders fifth edition (DSM-5) as a condition warranting further research [6], and more recently included in the 11th International Classification of Diseases (ICD-11) by the World Health Organization (WHO) under the term “Gaming Disorder” [7,8]”

  1. In the methodology section, I believe there needs to be more information about the "forum guilds" that were used to recruit participants. For instance, what is a forum guild, and why was a forum guild used? Provide more detail for what MMORPGs are and why they were used in participant recruitment. Regarding the adapted measure for IGD, there needs to be more background regarding how it is used for measurement. In addition, why was this measure used over others given that this is a rather novel construct? Lastly, why were the other psychological assessments utilized over other measurements? Please provide some more rationale for why you chose your measurements.

As request the reviewer, we added information about forum guilds in the methodological section.

Line 564-574

“MMORPG gamers with IGD, were selected from a French cohort recruited in our online study called Add MMORe, and constituted the MMORPG-IGD group. A cutoff ≥3 on the 7 items of the DSM-IV-TR substance dependence adapted scale (DAS) [48] was applied to determine in this MMORPG sample those qualifying for IGD. Add MMORe is an anonymous online study which recruited French MMORPG non-treatment seeking gamers from MMORPGs’ guild forums (private forums of gamers’ groups called guilds in WoW game where gamer's interact and discuss about gaming rules and tactic), between May 2009 and March 2010. All subjects gave their consent through electronic signature before starting the questionnaire. More information about the study design and descriptive results (socio demographic data, Internet addiction and gaming disorder screening tools and excessive gaming consequences) have previously been published [48].”

In addition, we added information about the DAS, scale used and developed by our team to evaluated the addictive disorder and more specifically developed for the assessment of the IGD in the MMORPG gamers. 

Line 584-593

“2.2.1. Dysfunctional engagement in MMORPGs, DSM IV-TR for substance dependence Adapted Scale (DAS)

DAS used in this study has been published previously as a good discriminating screening tool for addiction to MMORPGs [48]. It consisted In seven items (adapted from substance dependence DSM IV-TR criteria) related to online game MMORPG use over a past period of 12-months, answered « yes » or « no ». A threshold of ≥3 positive criteria on the 7 items was applied to determine whether MMORPG players qualified for IGD. In fact, at the time the study took place (2009-2010), IGD was not yet included in the DSM and the DSM-IV-TR criterion for addiction was the standard for assessing addictive disorders.”

  1. Lastly, why were the other psychological assessments utilized over other measurements? Please provide some more rationale for why you chose your measurements.

To answer at this question, we added in the introduction section few information about the self-report choice.

Line 173-175

“To assess alexithymia, the Toronto Alexithymia Scale (TAS-20) is commonly used [28], with a significantly higher score in the addicted population compared to healthy controls [23,24]”

Line 180-181

“Impulsivity is largely assessed by the Barratt Impulsivity Scale (BIS-10),  in various addicted samples [30].”

Line 184-185

“Usually, sensation seeking is measured by the sensation seeking scale (SSS-V) developed by Zuckerman [37].”

Line 202-204

“In this context, Buss-Durkee Hostility Inventory (BDHI), a validated self-administered questionnaire, appears as a simple way to assess the different kinds of hostility [47]. “

Round 2

Reviewer 2 Report

I think the paper has improved considerably and I accept it in present form.